# Live-cell imaging in human colonic monolayers reveals ERK waves limit the stem cell compartment to maintain epithelial homeostasis

Kelvin W Pond[1,2,3], Julia M Morris[1], Olga Alkhimenok[2], Reeba P Varghese[1,4], Carly R Cabel[1,4], Nathan A Ellis[1,3], Jayati Chakrabarti[1], Yana Zavros[1,3], Juanita L Merchant[5], Curtis A Thorne[1,3]*, Andrew L Paek[2]*

[1]Department of Cellular and Molecular Medicine, University of Arizona, Tucson, United States; [2]Department of Molecular and Cellular Biology, The University of Arizona, Tucson, United States; [3]University of Arizona Cancer Center, Tucson, United States; [4]Cancer Biology Graduate Interdisciplinary Program, University of Arizona, Tucson, United States; [5]Department of Medicine, The University of Arizona, Tucson, United States

*For correspondence:
curtisthorne@arizona.edu (CAT);
apaek@email.arizona.edu (ALP)

**Competing interest:** The authors declare that no competing interests exist.

**Abstract** The establishment and maintenance of different cellular compartments in tissues is a universal requirement across all metazoans. Maintaining the correct ratio of cell types in time and space allows tissues to form patterned compartments and perform complex functions. Patterning is especially evident in the human colon, where tissue homeostasis is maintained by stem cells in crypt structures that balance proliferation and differentiation. Here, we developed a human 2D patient derived organoid screening platform to study tissue patterning and kinase pathway dynamics in single cells. Using this system, we discovered that waves of ERK signaling induced by apoptotic cells play a critical role in maintaining tissue patterning and homeostasis. If ERK is activated acutely across all cells instead of in wave-like patterns, then tissue patterning and stem cells are lost. Conversely, if ERK activity is inhibited, then stem cells become unrestricted and expand dramatically. This work demonstrates that the colonic epithelium requires coordinated ERK signaling dynamics to maintain patterning and tissue homeostasis. Our work reveals how ERK can antagonize stem cells while supporting cell replacement and the function of the gut.

## Editor's evaluation

This work employs timelapse microscopy to study behavior of human colonic organoids in monolayers as the organoids initially self-organize. The authors then follow maintenance of organization into densely clustered nodes that have increased cells in cell cycle and the remaining more sparsely populated regions with fewer cycling cells. This study builds on a literature demonstrating roles for signaling pathways like ERK in epithelial patterning that have been examined in the cell competition field and, more specifically, in mouse intestinal organoids. This manuscript should be relevant to a broad readership interested in how epithelial organoids can self-organize and the role of specific signaling pathways in the process. In addition, the technical aspects of the work with live human monolayer cultures observed over timelapse with extensive quantification of cell behavior represent a useful advance in the field.

## Introduction

The colonic epithelium is a highly organized and rapidly renewing tissue containing spatially separated compartments called crypts. Roughly 10 million evenly spaced crypts inhabit the colon and are renewed every 3–5 days (*Zhao and Michor, 2013*). Each colonic crypt is ~2000 single cell monolayer invagination, roughly 82 cells in height and 41 cells in diameter (*Potten et al., 1992*). Cell extrusion and death are largely confined to the non-crypt compartments of the colon, while cell division, which is required to replace differentiated cell types, is restricted to the crypt base. The distance separating the stem cell and differentiated cell compartments is ~300 microns (*Sugimoto et al., 2018*; *Nguyen et al., 2015*). Chemical destruction of these millions of repeated structures can be repaired in the normal gut within weeks (*Lee et al., 2020*); thereby, re-establishing spatial separation between stem and differentiated cell compartments with remarkable accuracy. Across these cell compartments, the Wnt and MAPK pathways oppose one another to maintain homeostasis by balancing differentiation and stemness (*Tripurani et al., 2018*; *Wei et al., 2020*; *Kabiri et al., 2018*; *Ashton et al., 2010*; *Warzych et al., 2020*). ERK activity is low in colonic crypts and increases as cells become more differentiated. Wnt inhibition leads to hyperactive ERK signaling (*Kabiri et al., 2018*), suggesting that Wnt is suppressing ERK signaling in order to maintain stemness. Conversely, loss of ERK1/2 triggers the expansion of stem cells (*Wei et al., 2020*), suggesting ERK activity limits the stem cell pool. While much is known about the pathways required for maintaining the different cell types of the colon, how these compartments communicate with each other and maintain regular spacing is unknown.

MAPK/ERK signaling dynamics in wound healing have recently been described. Elegant work in Madin-Darby canine kidney (MDCK) cells found that cells at the wound barrier initiate ERK signaling cascades which propagate in wave-like patterns across the cell monolayer away from the site of damage. These ERK waves drive collective movement in the direction of the wound to efficiently seal barriers (*Matsubayashi et al., 2004*). ERK waves during wound healing have been observed in vitro and in vivo in a diverse set of animal models, suggesting a highly conserved function in the wound healing process. ERK waves are induced in response to apoptosis (*Gagliardi et al., 2020*) and promote hypertrophy (*De Simone et al., 2021*). Importantly, ERK dynamics can instruct fate decisions. In neuronal cells and *Drosophila* embryos, ERK signaling promotes differentiation if the signal is prolonged or total strength is increased (*Santos et al., 2007*; *Johnson and Toettcher, 2019*). A recent study using organoid monolayers derived from murine small intestine showed that differentiated villus cells have higher ERK signaling over time compared to crypt cells (*Pokrass et al., 2020*). In summary, ERK signaling can propagate across the epithelium in wave-like patterns in response to cues including apoptosis, regenerative need, and oncogenic mutations (*Gagliardi et al., 2020*; *De Simone et al., 2021*; *Hiratsuka et al., 2015*; *Aikin et al., 2020*). Yet, it is unclear whether ERK waves play a role in maintaining the proper ratios of cell types in the human colonic epithelium.

Herein, using self-organizing primary human colon organoid monolayers, we show that epithelial cells can reorganize from single cells into a complex monolayer resembling the tissue of origin. Regularly spaced crypt-like structures (hereafter referred to as 'nodes') form 3–4 days after cell dissociation and seeding. These nodes are maintained for weeks in spatially distinct compartments. In non-node compartments, cell extrusion and apoptosis are increased. As a cell is extruded and undergoes apoptosis, an ERK wave is initiated that propagates radially from the dying cell. Locally, cells in the path of the wave are prompted to migrate toward the dying cell, presumably to maintain epithelial integrity as reported recently (*Gagliardi et al., 2020*; *Valon et al., 2020*). The spacing of nodes is similar to the diameter of ERK waves, which suggests that the length scale observed between nodes is dependent on ERK waves. Consistent with this, we found that if ERK activity is globally increased, nodes disappear and cease to express the Wnt signaling and stem cell markers *LGR5* and *MYC*. Alternatively, if ERK is inhibited globally, nodes increase in size and express high levels of *LGR5* and *MYC*. If Wnt3a is removed from culture media, nodes also disappear and ERK activity is increased. These data show that Wnt and ERK mutually oppose one another to maintain gut homeostasis and Wnt-suppressing ERK waves originating from apoptotic cells help to shape the architecture of the human colonic epithelium.

## Results

### Patient-derived colonic organoid monolayers

To study single-cell dynamics in the human colon, we adapted our previous method of culturing murine intestinal organoids as monolayers. These monolayers maintain patterning and cell fate characteristics observed in 3D organoid systems and in vivo (*Thorne et al., 2018*). In conjunction with the University of Arizona Cancer Center organoid resource (BioDROid), we collected colonic normal, adenomatous polyp, and carcinoma samples. We cultured them as 3D patient-derived organoids, expanded them in 3D and transferred them onto a thin layer of extracellular matrix after dissociation into single cells. After 4 days, confluent monolayers harbored phenotypically distinct clusters and were considered fully established (*Figure 1A and C*). Strikingly, even starting as a random dispersion of single cells, normal colonic organoids developed localized compartments of densely packed cells (nodes) that were surrounded by less dense cells. These nodes resembled those we observed using the murine model system (*Thorne et al., 2018*), with LGR5 +stem cells located specifically within densely compacted cell compartments (*Figure 1—figure supplement 1A*). Murine small intestinal organoid monolayers and normal human colon monolayers rarely reached confluence and instead reached a homeostatic state and maintain nodes at subconfluency (*Figure 1—figure supplement 1A-C*). A polyp-derived 2D organoid line (referred here as GiLA1 for Gastrointestinal Line, Arizona 1) formed regularly spaced nodes, quickly reached confluence within 72 hr, and could maintain node structures for 45 days or more. When GiLA1 single cells were seeded onto an ultra-thin layer of Matrigel, organoids formed monolayers by 3 days and began to spontaneously form regularly spaced compact compartments reminiscent of colonic crypts within 3–4 days (*Figure 1C and D*). A tumor organoid derived from a micro satellite stable (MSS) invasive colon adenocarcinoma (p21T) was grown in 2D and displayed different morphologies compared to normal, polyp, and the murine small intestine (*Figure 1—figure supplement 1B*). p21T proliferated but failed to form nodes and instead formed uniform, disorganized monolayers (*Figure 1—figure supplement 1B*). To date, across multiple patient-derived organoid lines, we have observed the formation of nodes in normal and polyp-derived organoids; however, we fail to observe nodes in multiple adenocarcinoma organoids. Across five colorectal patient lines obtained from the PDMR, no patterning was observed, and all lines formed confluent monolayers. (*Figure 1—figure supplement 1D*). Because of its unique ability to form both complete monolayers and distinct cellular compartments, the GiLA1 organoid line was used for the remainder of the described experiments.

### Self-organization of organoid monolayers

A striking aspect of the organoid monolayers is that they appear to self-organize into node/non-node compartments. To capture the establishment of these structures over time, we used time-lapse microscopy of GiLA1 organoid monolayers expressing H2B-iRFP670 to observe the formation of distinct cellular compartments over 5 days. Cells initially exhibited high levels of motility after attachment to the extracellular matrix (ECM) to form 5–10 cell clusters. After this, cells became hyperproliferative from 48 to 72 hr after seeding to form a complete monolayer. Finally, morphologically distinct cellular compartments were formed by 96 hr (*Figure 1E*, *Figure 1—video 1*). These 2D monolayers can be readily grown in 384-well imaging plates making this model particularly amenable to high-content imaging and quantitative-image analysis. Together, these data show that patient biopsies from the human colon can fully self-organize into patterned monolayers that maintain long-term tissue homeostasis.

### Spatially distinct stem and differentiated cell compartments

We further characterized the GiLA1 organoid line by determining the cell types in organoid monolayers and their location with respect to the nodes. Two proliferative markers, Ki-67 and EdU, revealed that proliferative cells were much more abundant within the compartments of densely packed cells of the organoid monolayers (*Figure 2A and C*), suggesting the nodes are a collection of transit amplifying-like cells and possibly stem cells. To determine if nodes harbored stem cells, we performed single-molecule RNA fluorescence in situ hybridization (smRNA FISH) against Wnt target genes and stem cell markers, LGR5 and MYC. *LGR5* and *MYC* expressed at higher levels in nodes compared to non-nodes (*Figure 2B and D*). Cells expressing the differentiated-colonocyte marker *KRT20* were primarily located in non-node regions and were mutually exclusive from LGR5 high compartments

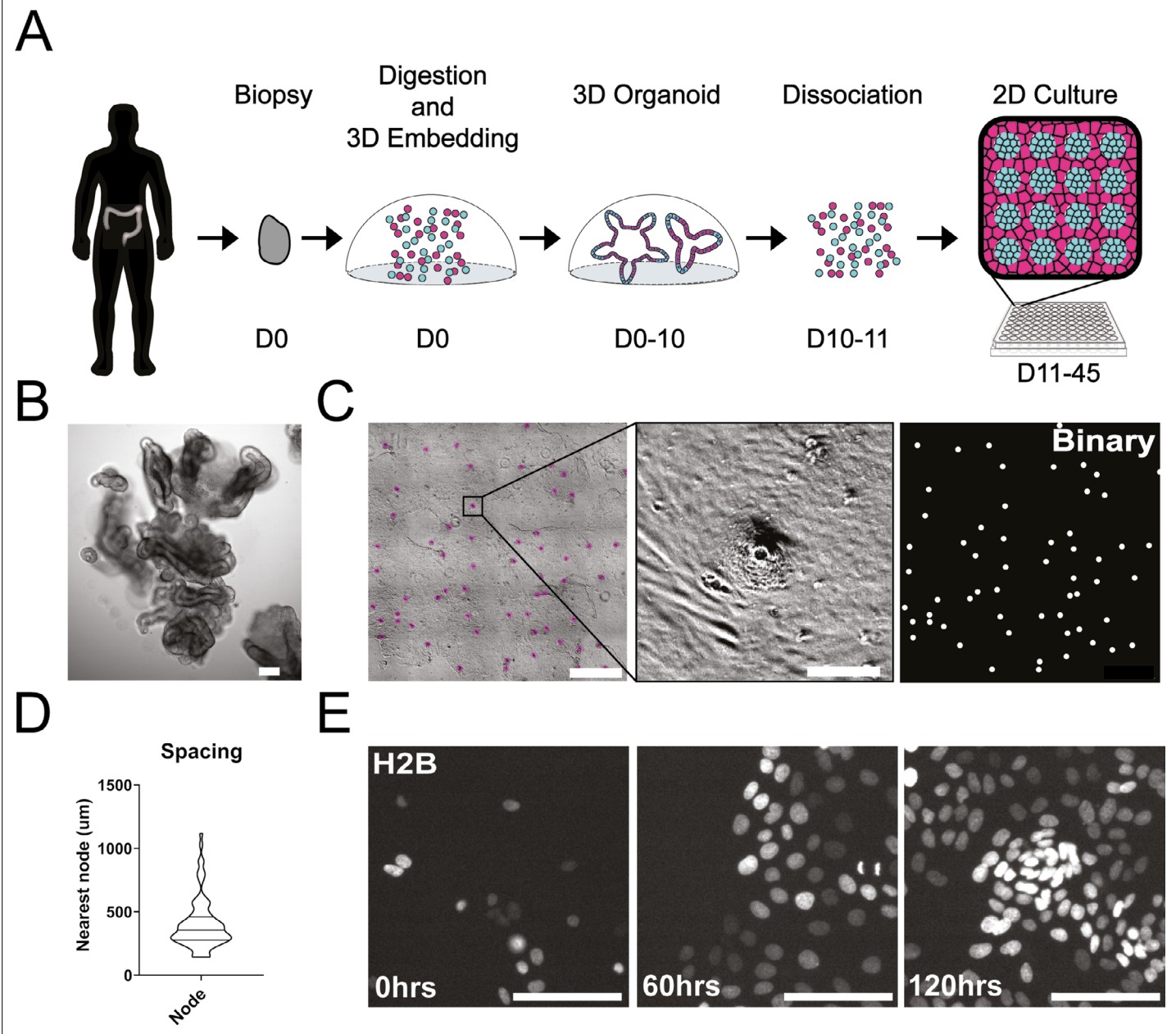

**Figure 1.** Development and self-organization of colonic patient-derived colonic organoid monolayers. (**A**) Model depicting the workflow for the development of organoid monolayers. Patient biopsies are digested using collagenase to form a single cell suspension. Cells are then embedded in Matrigel and grown in 3D using a defined organoid growth medium. Organoids are expanded in 3D before seeding onto 2D imaging plates coated with a thin layer of Matrigel. Within 5 days, single cells self-organize to form a regularly patterned organoid monolayer. Organoid monolayers maintain homeostasis in this patterned state for up to 45 days. (**B**) Representative brightfield image of a single 3D colonic organoid. (**C**) Representative brightfield image of organoid monolayer. Left-automated image segmentation of nodes across a single well. Automated detection of nodes is shown in purple. Middle-sSingle node, zoom of left. Right-binary image of segmentation showing regular spacing of nodes across the culture. (**D**) Violin plot showing the distribution of spacing across organoid monolayers. Data represents quantification of 84 nodes (*Figure 1—source data 1*). (**E**) Live cell images of self-organization over 5 days using organoids expressing H2B-iRFP670 for nuclear tracking. All scale bars represent 100 µM except C, left, which represents 1000 µM.

The online version of this article includes the following video, source data, and figure supplement(s) for figure 1:

**Source data 1.** Source data for *Figure 1D*.

**Figure supplement 1.** Organoid monolayer patterning in normal and tumor- derived tissues.

**Figure 1—video 1.** Self-organization of 2D Monolayers.

https://elifesciences.org/articles/78837/figures#fig1video1

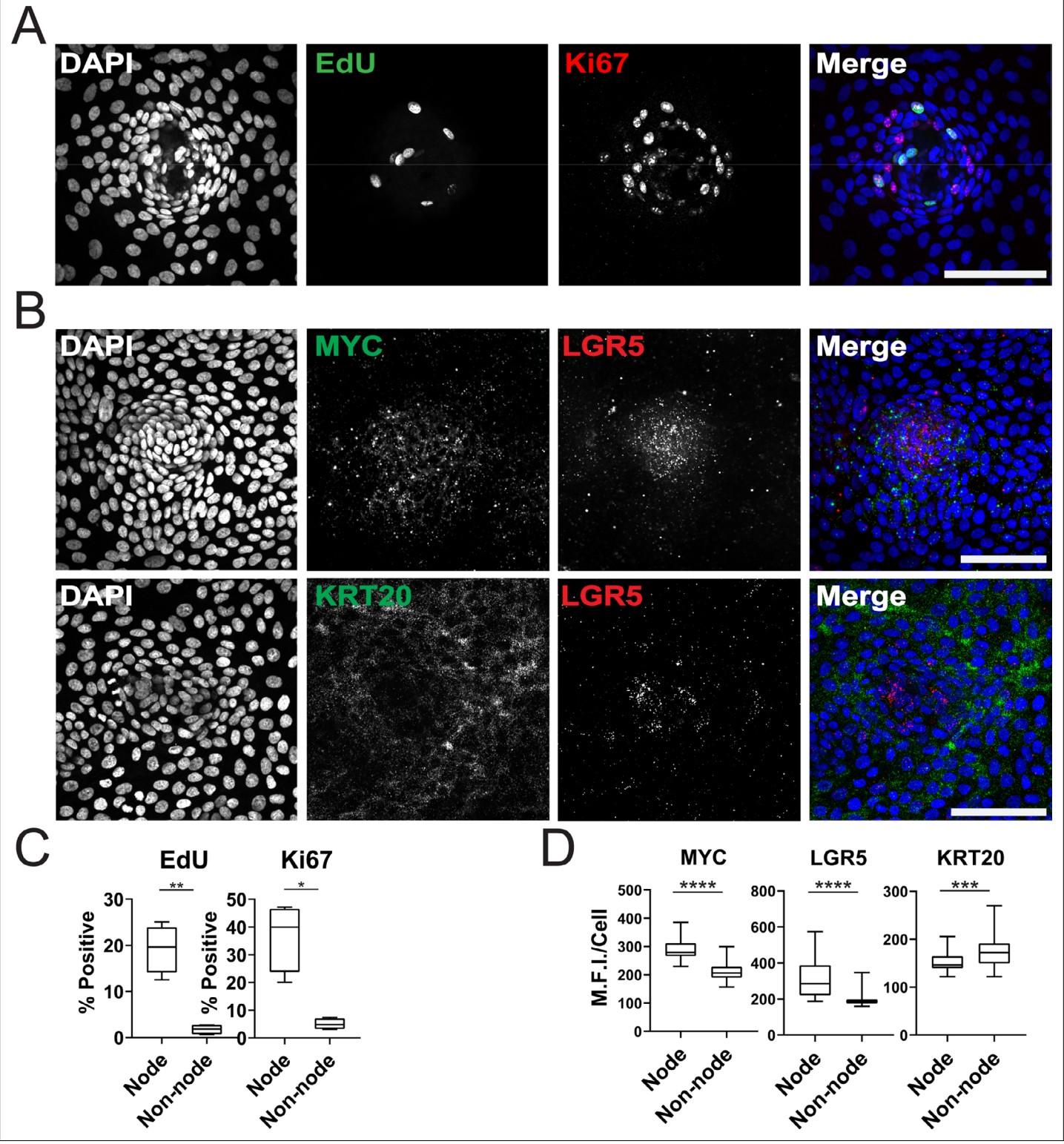

**Figure 2.** Characterization of Organoid Monolayer Compartments. (**A**) Representative *Figure 2D* organoids stained for nuclei (blue), EdU (green), and ki67 (red). (**B**) Representative images of single-cell RNA fluorescence in situ hybridization (FISH) against Wnt target genes MYC and LGR5 in a single node. (**C**) Quantification of A showing the percent of positive cells in node or non-node regions, asterisks represent significance from paired t-test (*Figure 2—source data 1–2*). (**D**) Quantification of images represented in B, mean fluorescence intensity (MFI) of target transcripts in each single cell is shown. Cells were separated based on presence in nodes vs non-nodes. Three to four biological replicates were performed. Data shown is from

*Figure 2 continued on next page*

Figure 2 continued

analysis of 64–224 cells binned from 16 technical replicates (*Figure 2—source data 3–5*). Asterisks represent significance from Mann Whitney analysis, ****=p<0.0001 etc. All scale bars represent 100 µM.

The online version of this article includes the following source data for figure 2:

**Source data 1.** Source data for *Figure 2C*_EdU.

**Source data 2.** Source data for *Figure 2C*_ki67.

**Source data 3.** Source data for *Figure 2D*_MYC.

**Source data 4.** Source data for *Figure 2D*_LGR5.

**Source data 5.** Source data for *Figure 2D*_KRT20.

(*Figure 2B and D*). Organoids harbored a majority of colonocytes, whereas tuft, goblet, stem, and proliferative cell markers were observed at low levels indicating normal cell fates were maintained in GiLA1 monolayers (*Figure 1—figure supplement 1E and F*). These data show that patterned organoid monolayers contain cells positive for stem, proliferative, and differentiated cell markers.

## Apoptosis-induced ERK waves instruct cell movement

MAPK/ERK signaling is a critical regulatory pathway involved in maintaining homeostasis of epithelia in the gut (*Wei et al., 2020*). To assess ERK activity in human organoid monolayers over time, we utilized a well-characterized ERK kinase translocation reporter (ERK-KTR) (*Regot et al., 2014*). The ERK-KTR utilizes a bipartite nuclear localization signal (NLS) and nuclear export signal (NES) to transform ERK kinase activity into a nuclear to cytoplasmic shuttling event that is easily quantified by microscopy in organoid monolayers (*Figure 3A*). We transduced the GiLA1 organoid line with a H2B-iRFP670 nuclear marker and ERK-KTR-mRuby2 reporter to perform nuclear and cytoplasmic segmentation and calculate nuclear-to-cytoplasmic ratios of the ERK-KTR (*Figure 3A*, *Figure 3B* and *Figure 3—figure supplement 1A*). In fully developed organoid monolayers, we observed a~twofold reduction of the ERK-KTR after treatment with inhibitors of EGFR or MEK across all cell types, suggesting the KTR is actively representing ERK signaling and is EGFR dependent (*Figure 3—figure supplement 1B*). Interestingly, when GiLA1 monolayers were imaged using time lapse microscopy, we observed ERK activation originating at focal points within the confluent monolayer that propagate outward in a wave-like pattern. These waves did not overlap and were observed routinely during homeostasis (*Figure 3C*, *Figure 3—video 1*). Imaging of monolayers using brightfield and caspase 3 dye revealed apoptotic cells that were being extruded from the monolayer to be the focal source of wave activation (*Figure 3—video 1*).

To determine the properties of the ERK waves, we tracked ~450 cells and assessed their ERK activity over time following an apoptotic event (*Figure 3D*, left panel and *Figure 3E*). During the apoptotic event, the cells immediately adjacent to the apoptotic cell moved away from the apoptotic cell slightly. The apoptotic event was followed by a wave of ERK activity that traveled approximately 450 µM over a period of 95 min for an average speed of ~4.7 µM/min. In addition to the ERK signaling wave, we noticed motility of the cells surrounding the apoptotic cell towards the site of apoptosis. Tracking single cells revealed a striking correlation between intensity/duration of ERK activity and movement of cells surrounding the apoptotic cell in the direction of the apoptotic cell (*Figure 3D*, two right panels and 3E). As the ERK wave propagated outward from apoptotic cells, it dissipated in both duration and intensity (*Figure 3F*). Particle image velocimetry (PIV), a method used to quantify the flow of image pixels over time (*Adrian, 1991*), confirmed collective cell migration towards the apoptotic cell (*Figure 3G*), suggesting that ERK waves instruct migration of surrounding cells as shown in previous studies (*Gagliardi et al., 2020*). Cells treated with the EGFR inhibitor Gefitinib did not induce ERK waves or display tropism towards dying cells, suggesting that EGFR-mediated ERK signaling is required for cell movement in this context (*Figure 3G*, *Figure 3—video 2* and *Figure 3—video 3*). Tracking of single cells (*Figure 3E*) and cross correlation analysis (*Figure 3—figure supplement 2*) revealed that ERK activity preceded cell movement, suggesting that ERK waves instruct cell movement. We previously described distinct apoptotic and proliferative compartments in organoid monolayers derived from murine small intestine (*Thorne et al., 2018*). To confirm this in human organoid monolayers and investigate if cell death was restricted to a compartment distinct from proliferative nodes, we imaged mature living organoid monolayers in the presence of cleaved caspase 3 dye, a

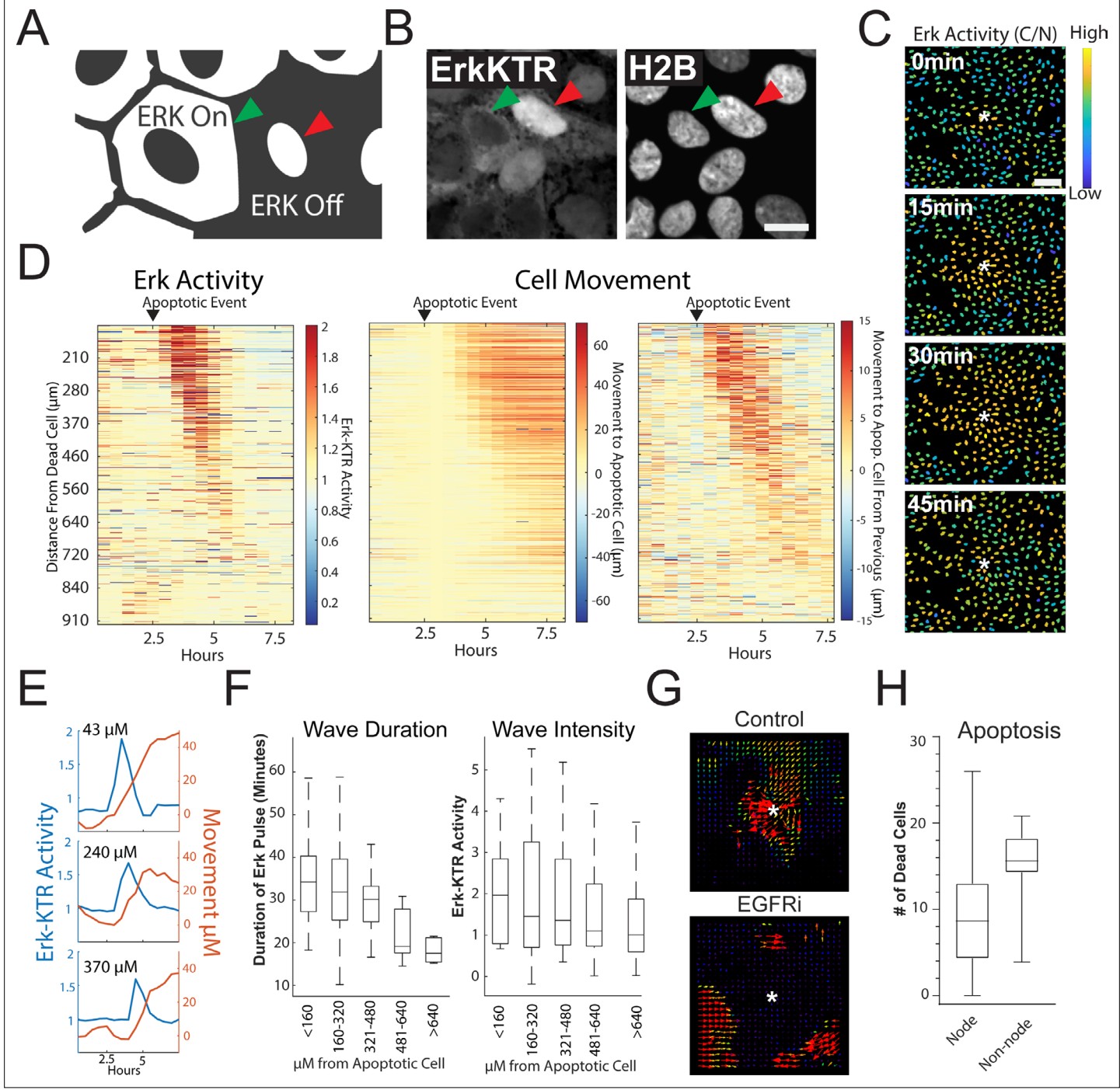

**Figure 3.** Apoptosis Induces an ERK Signaling Wave that Instructs Cell Movement. (**A**) Model depicting how to interpret the ERK-KTR translocation reporter. Cells with mostly cytoplasmic ERK-KTR have high ERK activity (left, green arrow). Cells with mostly nuclear ERK-KTR signal have low ERK activity (right, red arrow). Scale bar represents 10 μM. (**B**) Representative images of the ERK-KTR active vs inactive. (**C**) Representative images showing a single ERK wave propagating from an apoptotic cell (white*). Heat map of the nuclear to cytoplasmic ratio of the ERK-KTR is shown in blue (low)/yellow (high). Scale bar in top image represents 100 μM. (**D**) Left: single-cell analysis of ERK activity over time after an apoptotic event. Heat maps are ordered from closest (top) to furthest (bottom) distance from the dying cell over time. ERK activity is shown (red, high ERK activity; blue, low ERK activity). Middle-single cell analysis of cell movement obtained from the same dataset shown on left. Change in distance towards the position of the apoptotic event is shown. Right-relative change in distance compared to the previous frame is shown. Data is represented from a single wave event. 455 cells were analyzed. (**E**) Representative single-cell traces comparing ERK activity (blue) and cell movement (red) at a given distance from the apoptotic cell is shown. (**F**) Left: duration of ERK signaling wave at a given distance from an apoptotic cell. Right- ERK-KTR activity at a given distance from an apoptotic cell (*Figure 3—source data 1*). (**G**) Representative images of particle image velocimetry (PIV) of H2B-iRFP[670] images after an apoptotic event. Organoids

*Figure 3 continued on next page*

*Figure 3 continued*

were treated with or without 5 µM Gefitinib. These data correspond with movie 3. Arrows indicate direction and amplitude of movement for the duration of the movie. Asterisk represent the position of the apoptotic cell. (**H**) Location of cleaved caspase three positive cells in monolayer over 24 hr. Data was acquired from 96 time points and is represented as the number of caspase positive cells in node or non-node area (*Figure 3—source data 2*).

The online version of this article includes the following video, source data, and figure supplement(s) for figure 3:

**Source data 1.** Source data for *Figure 3D–F*.

**Source data 2.** Source data for *Figure 3H*.

**Figure supplement 1.** Generation and validation of ErkKTR Organoids from Human Biopsies.

**Figure supplement 2.** Cross-correlation Analysis of Cell Movement vs ERK Activity.

**Figure supplement 3.** Representative Images of Caspase 3 Dye Localization in node vs non-node areas.

**Figure 3—video 1.** ERK Waves Originate from Apoptotic Cells.
https://elifesciences.org/articles/78837/figures#fig3video1

**Figure 3—video 2.** Cell Movement is Instructed by an ERK wave.
https://elifesciences.org/articles/78837/figures#fig3video2

**Figure 3—video 3.** Cell Movement is Instructed by an ERK Wave.
https://elifesciences.org/articles/78837/figures#fig3video3

marker for apoptosis. We found that the emergence of an apoptotic cell was more likely in differentiated cells in non-node compartments (*Figure 3H* and *Figure 3—figure supplement 3*). Taken together, these data show that ERK waves induced by apoptotic cells instruct surrounding cells to migrate toward the apoptotic site.

## ERK waves instruct patterning in organoid monolayers

Increased ERK kinase activity has been reported in differentiated cells compared to the stem cell compartment within the gut (*Kabiri et al., 2018*). We asked if ERK activity was increased in the differentiated compartment in GiLA1 monolayers. To determine this, we created a time lapse movie of a dense node compartment surrounded by less dense, non-node cells over a 16 hr period. Nuclear to cytoplasmic ratio of the ERK-KTR was measured at the single cell level over time. A threshold for active ERK cells was then combined with spatial clustering to identify ERK-active regions. High ERK regions were then superimposed from each imaging timepoint to create a single heat map for regional ERK activity over time. Consistent with prior studies using murine small intestine (*Kabiri et al., 2018*), cells in non-node compartments were more mobile and displayed more active ERK signaling (*Figure 4A* and *Figure 4—video 1* and *Figure 4—video 2*). Together, these data show that apoptotic events that cause ERK waves are mostly localized in spatially distinct differentiated cell compartments.

Since ERK waves are largely restricted to the postmitotic, differentiated compartments in our gut organoid monolayer model, we hypothesized that ERK waves may help regulate the patterning observed in the GiLA1 monolayers by restricting the stem cell compartment. To test this hypothesis, we asked if global activation of ERK activity could disrupt the patterning of the organoid monolayers. We used a well-established activator of the ERK pathway**,** phorbol 12-myristate 13-acetate (PMA) to acutely activate ERK signaling across the monolayer. Following treatment with 100 nM PMA, we observed immediate hyperactivated ERK in all cells (*Figure 4B and C*, *Figure 4—video 3*). Monolayers showed a remarkable loss of patterning after PMA treatment as evidenced by almost complete loss of cell clusters by 24 hr post-treatment (*Figure 4D–F*). To determine if nodes re-formed once ERK activity returned to baseline, we treated GiLA1 monolayers with a pulse of 100 nM PMA for 1 min. The pulse PMA treatment caused ERK activation in all cells after 30 min, followed by node loss. Notably, ERK levels returned to normal after PMA treatment, but nodes did not re-form by 48 hr (*Figure 4G* and *Figure 4—video 4*). These data suggest a potential role for a ERK waves in regulating the ratio of stem to differentiated cells in organoid monolayers.

In support of this hypothesis, we observed a correlation between the average distance (~399 µm) between neighboring proliferative compartments and the average distance (~447 µm) covered by the apoptosis-induced ERK wave (*Figure 4—figure supplement 1A*). This result was maintained across multiple biological replicates with high fidelity (*Figure 4—figure supplement 1B*).

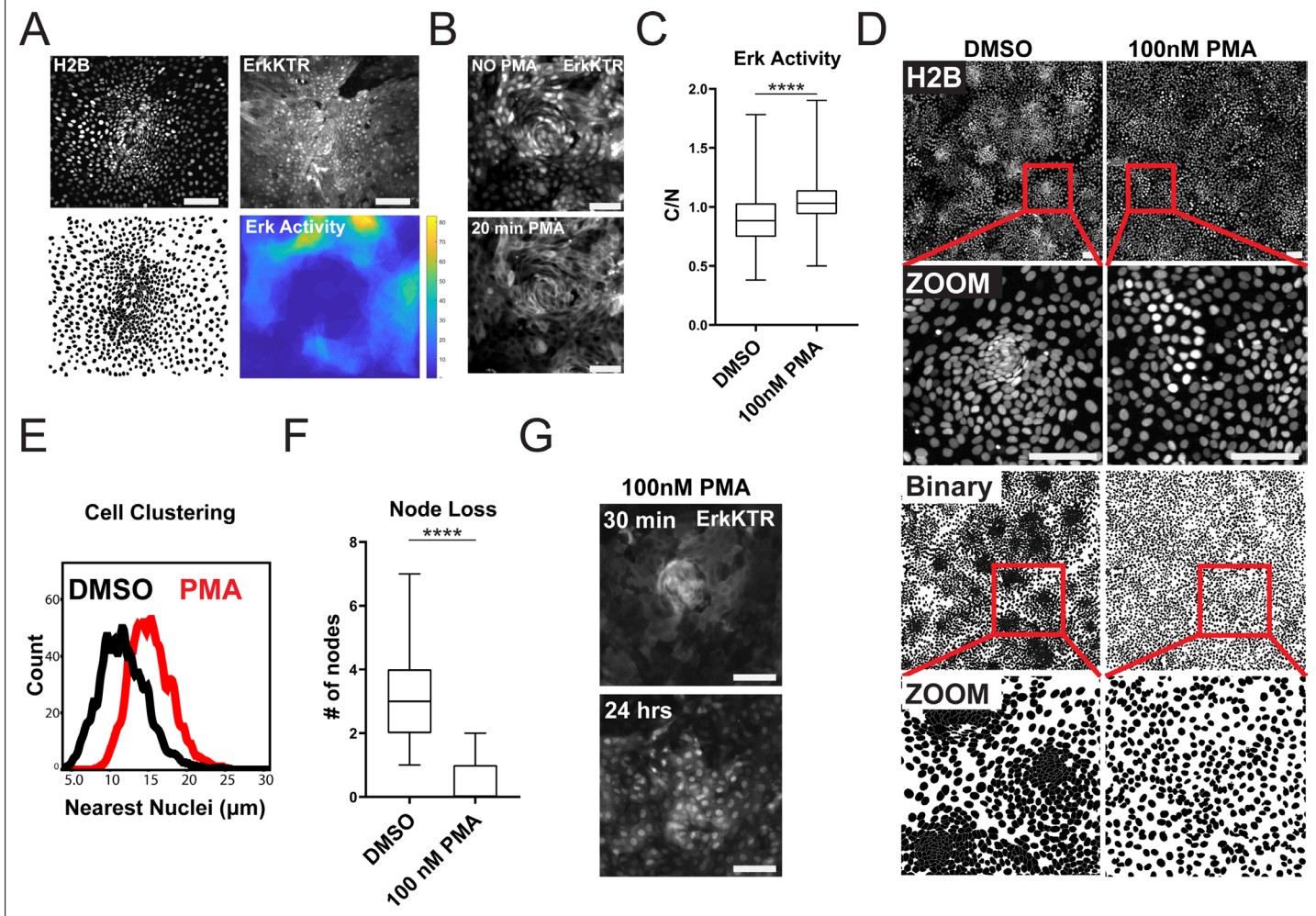

**Figure 4.** ERK Dynamics are Essential to Maintain Tissue Patterning in Organoid Monolayers. (**A**) Representative images of cell density variation in node vs non-node. Top left: H2B-iRFP670. Top right: ERK-KTR. Bottom left- binary nuclear segmentation showing high- and low-density regions. Bottom right: a spatial heat map of ERK activity over 16 hr in node and surrounding region. (**B**) Representative images of ERK activation after 100 nM phorbol 12-myristate 13-acetate (PMA) treatment for 20 min. Top- ERK-KTR before treatment with PMA. Bottom- ERK-KTR of the same node 20 min after PMA treatment. (**C**) Quantification of ERK-KTR activity after 30 min of 100 nM PMA treatment. Analysis from 255 single cells is shown (*Figure 4—source data 1*). (**D**) Representative images of cell clustering following 100 nM PMA treatment. Top 4 panels show nuclear distribution across stitched image (top) and zoom (bottom) before and after PMA treatment. Bottom 4 panels show binary segmentation to clearly show loss of cell clustering following PMA treatment. (**E**) Histograms showing distance between nuclei with or without treatment with 100 nM PMA. Data is represented as nearest nuclei distance. Analysis of 3523 cells is shown (*Figure 4—source data 2*). (**F**) Quantification of patterning loss before and after treatment with 100 nM PMA for 24 hr. Analysis of 53 images is shown (*Figure 4—source data 3*). (**G**) Representative images of ERK-KTR activity after pulsing cells for 1 min with PMA followed by washout and chase for 24 hr. Top- all cells show high ERK activity with intact node. Bottom-node loss and resuppression of ERK activity after 24 hr. Asterisks represent significance from Mann Whitney analysis, ****=p<0.0001 etc. Scale bars represent 100 μM.

The online version of this article includes the following video, source data, and figure supplement(s) for figure 4:

**Source data 1.** Source data for *Figure 4C*.

**Source data 2.** Source data for *Figure 4E*.

**Source data 3.** Source data for *Figure 4F*.

**Figure supplement 1.** Correlation Between Node Spacing and Wave Distance.

**Figure 4—video 1.** ERK Activity is Increased in Non-node Regions.

https://elifesciences.org/articles/78837/figures#fig4video1

**Figure 4—video 2.** Spatial Heat Map of ERK Activity Over Time of a Single Node.

https://elifesciences.org/articles/78837/figures#fig4video2

**Figure 4—video 3.** Activation of ERK-KTR using Phorbol 12-myristate 13-acetate (PMA).

*Figure 4 continued on next page*

*Figure 4 continued*

https://elifesciences.org/articles/78837/figures#fig4video3

**Figure 4—video 4.** 1 min Pulse of Phorbol 12-myristate 13-acetate (PMA) Induces Lasting Node Loss and Transient ERK Activation.

https://elifesciences.org/articles/78837/figures#fig4video4

PMA is a known activator of protein kinase C (PKC) and could change cellular pathways driven by PI3K, MMPs, FAK, and calcium (*Frost et al., 1994*; *Shih et al., 2010*; *Kim et al., 2010*). To test if genetic activation of ERK could drive patterning loss we used an established doxycycline-inducible KRAS$^{G12V}$ system (*Fuerer and Nusse, 2010*). Within 4 hr of treatment with doxycycline, GFP-tagged KRAS was induced in GiLA1 organoid monolayers and resulted in ERK activation after 7 hr (*Figure 5A–C*). Oncogenic KRAS activation can lead to hyperproliferation and genomic instability (*Haigis et al., 2008*; *Di Micco et al., 2006*). In line with this, GiLA1 monolayers displayed hyperproliferation and mitotic defects (genomic instability) after 16 hr of KRAS$^{G12V}$ expression (*Figure 5D–F*). Patterning was assessed visually each day and loss of nodes occurred markedly slower compared to PMA treatment. KRAS$^{G12V}$ induction resulted in a ~twofold reduction in nodes after 7 days (*Figure 5G–H*) and complete node loss after 14 days (*Figure 5—figure supplement 1*). If KRAS$^{G12V}$ was induced after organoids formed monolayers but before node formation (on day 2 after seeding), KRAS$^{G12V}$ expression led to a lack of node development in organoid monolayers (*Figure 5—video 1* and *Figure 5—video 2*). These data demonstrate that oncogene-driven activation of ERK can lead to patterning loss in organoid monolayers.

Wnt is a well-known promoter of stem cell renewal within the gut and may be responsible for the clustering of the proliferative nodes as we previously observed in murine small intestine monolayers (*Thorne et al., 2018*). We asked if removal of Wnt from growth conditions would affect stem cell numbers from our GiLA1 monolayers. Using smRNA FISH towards stem cell marker LGR5, we observed node structures were enriched in LGR5 + cells and these cells were reduced if Wnt3a was removed for 4 days from organoid culture media (*Figure 6A and B*). ERK activity was also significantly increased as cells became more differentiated following Wnt removal (*Figure 6C and D*, *Figure 6—video 1* and *Figure 6—video 2*), consistent with previous work showing increased ERK within crypts following Wnt inhibition (*Kabiri et al., 2018*). To confirm that loss of Wnt signaling can disrupt patterning and activate ERK, we treated 2D monolayers with the well-established Wnt inhibitor pyrvinium (*Thorne et al., 2010*; *Li et al., 2021*). Pyrvinium inhibited expression of Wnt target genes as measured by TCF/LEF-dependent GFP expression in GiLA1 monolayers after 72 hr (*Figure 6—figure supplement 1*). Consistent with the effect of Wnt removal, ERK activation could be seen within 4 hr in a dose dependent manner (*Figure 6E* - top right). Node loss was induced by treatment with pyrvinium in a dose dependent manner after 4 days (*Figure 6E* - bottom right). To measure the direct effects of either inhibition or activation of the ERK pathway across all cell types, we treated the GiLA1 monolayers with either a MEK inhibitor or PMA for 72 hr. Strikingly, blocking the ERK pathway increased the expression of LGR5 and resulted in larger clusters of densely packed cells, suggesting ERK waves limit stem cell expansion (*Figure 6F and G*). Conversely, treatment with PMA resulted in almost a total loss of stem cells and node structures (*Figure 6F and G*). Similar results were observed when measuring MYC transcripts in the same context (*Figure 6—figure supplement 2*). Collectively, these data support a model where ERK signaling, induced by cell death, promotes differentiation, and restricts stem cells to a defined compartment (*Figure 7*). Interrogation of either the Wnt or ERK pathways revealed a mutually antagonistic relationship between these central regulatory pathways which maintains homeostasis.

## Discussion
### ERK waves in the human gut

The human colon requires ongoing regeneration and renewal over a lifespan; this need relies on finely tuned communication between differentiated and stem cells. We show here that one way the colonic epithelium maintains both complexity and plasticity is through long-distance signaling driven by cell turnover. As cells differentiate and are eliminated from the epithelium, dying cells initiate an ERK wave that triggers local cell tropism towards the death event and stimulates differentiation of replacement

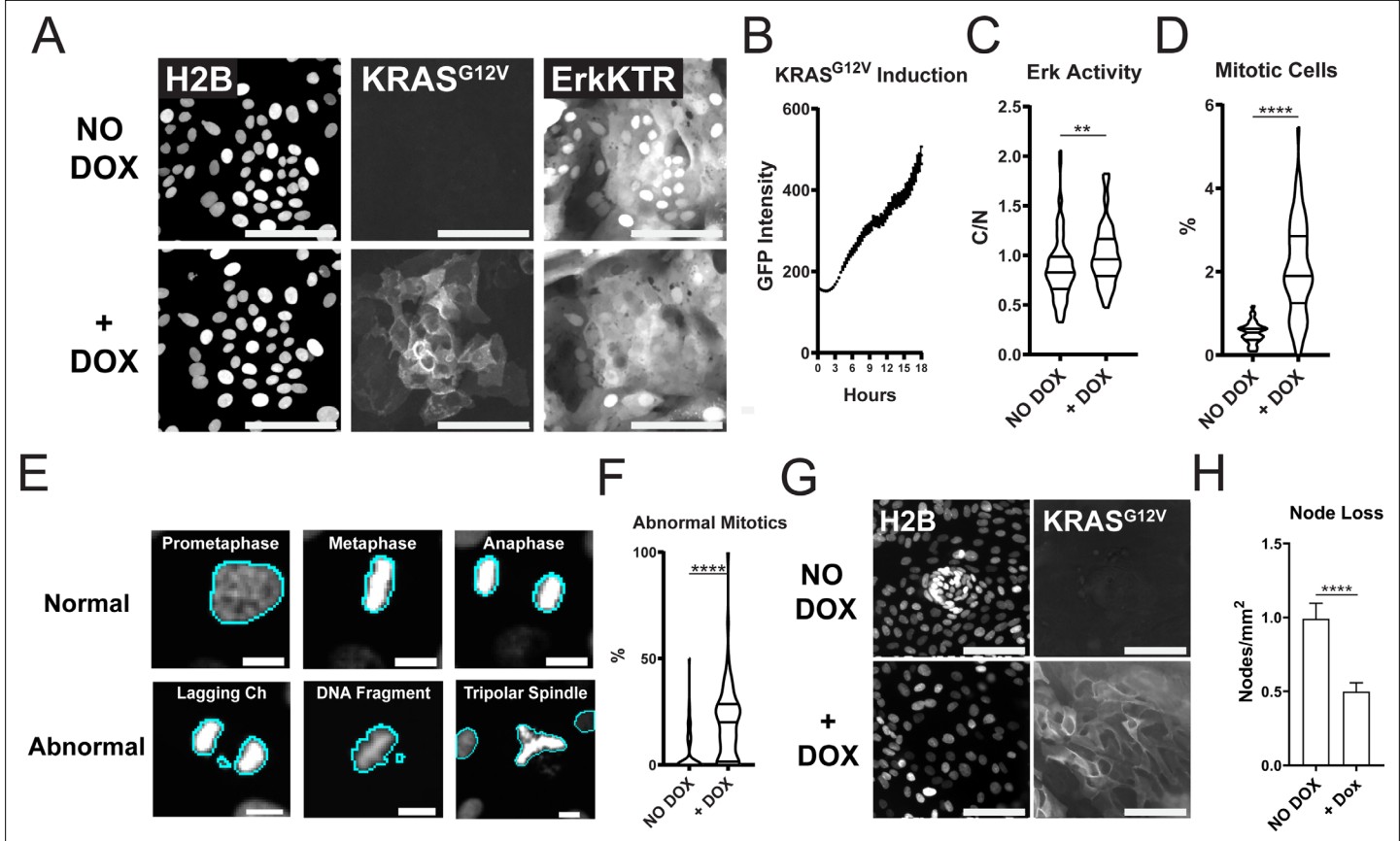

**Figure 5.** Expression of Oncogenic KRAS[G12V] Induces ERK Activation, Increased Proliferation, Mitotic Abnormalities, and Tissue Patterning Loss. (**A**) Representative images of GiLA1 monolayers expressing H2B, Dox-inducible GFP-KRAS[G12V], and ERK-KTR before (top) and 7 hr after (bottom) 1 µg/ml of doxycycline. Scale bars represent 100 µm. (**B**) Quantification of KRAS[G12V] induction after treatment with 1 µg/ml doxycycline for 16 hr. Analysis from 16 images over 54 timepoints is shown (**Figure 5—source data 1**). (**C**) Quantification of ERK-KTR before and after 7 hr of doxycycline. Data is represented as the cytoplasmic to nuclear ratio of the KTR intensity (**Figure 5—source data 2**). Analysis of 85–91 single cells is shown. (**D**) Quantification of the percent of mitotic cells with or without treatment with 1 µg/ml doxycycline for 16 hr (**Figure 5—source data 3**). (**E**) Representative images of normal or abnormal mitotic events taken from experiments described in figures C and D. Scale bars represent 10 µm. (**F**) Quantification of abnormal mitotic events in cells treated with 1 µg/ml doxycycline for 16 hr. Data is represented as percent of total mitotic cells within the image. At least 58 images were analyzed for each condition. Images harbored 400–1200 cells (**Figure 5—source data 4**). (**G**) Representative images of nodes in H2B and GFP-KRAS[G12V] expressing cells with or without 1 µg/ml doxycycline for 7 days. Scale bars represent 100 µm. (**H**) Quantification of images shown in G (**Figure 5—source data 5**). Data is represented as mean and SEM of at least 64 images harboring 400–1200 cells. Asterisks represent significance from Mann Whitney analysis, ****=p<0.0001 etc.

The online version of this article includes the following video, source data, and figure supplement(s) for figure 5:

**Source data 1.** Source data for **Figure 5B**.

**Source data 2.** Source data for **Figure 5C**.

**Source data 3.** Source data for **Figure 5D**.

**Source data 4.** Source data for **Figure 5F**.

**Source data 5.** Source data for **Figure 5H**.

**Figure supplement 1.** Loss of Patterning After KRAS[G12V] Induction.

**Figure 5—video 1.** Spontaneous Formation of Nodes After Monolayer Formation in Uninduced KRAS[G12V] Gila1 Monolayers.
https://elifesciences.org/articles/78837/figures#fig5video1

**Figure 5—video 2.** Lack of Node Formation in Gila1 Monolayers after KRAS[G12V] Induction by Doxycycline.
https://elifesciences.org/articles/78837/figures#fig5video2

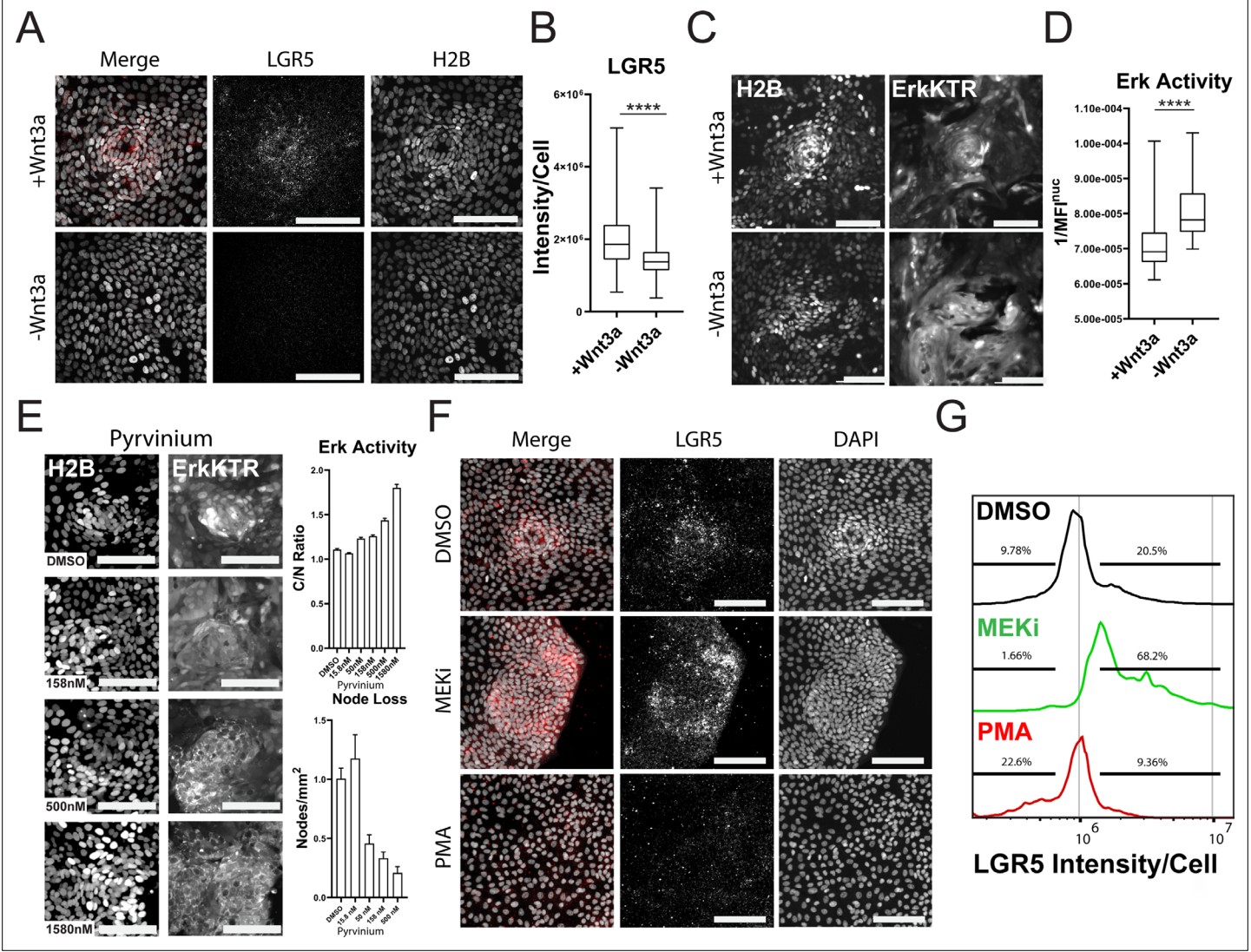

**Figure 6.** Wnt and ERK Signaling Mutually Limit Each Other to Preserve Tissue Homeostasis. (**A**) Representative images of LGR5 mRNA in organoid monolayer after removal of Wnt3a for 72 hr. H2B-GFP is shown in white and LGR5 mRNA is shown in red. (**B**) Quantification of scRNA FISH for LGR5 in organoid monolayers after removal of Wnt3a from organoid media. Data are represented as the sum of LGR5 intensity per cell. Analysis of at least 386 cells is shown (*Figure 6—source data 1*). (**C**) Representative images of ERK kinase activity after Wnt3a removal for 24 hr. Inverted nuclear intensity of ERK-KTR is shown. (**D**) Quantification of ERK kinase activity after removal of Wnt3a from organoid media from 0 to 48 hr following Wnt removal. Analysis of at least 287 cells is shown (*Figure 6—source data 2*). (**E**) Left: Representative images of H2B (left) and ERK-KTR (right) with or without treatment with a dose response of Pyrvinium for 4 hr. Top right: quantification of ERK activity after treatment with the Wnt inhibitor pyrvinium for 4 hr. Analysis of at least 1844 cells is shown. Bottom right-quantification of node loss after treatment with a dose response of pyrvinium for 4 days. Analysis of 14–79 images is shown (*Figure 6—source data 3* and *Figure 6—source data 4*). (**F**) Representative images of smRNA FISH for LGR5 in organoid monolayers after treatment with either 1 μM PD0325901 (MEKi) or 100 nM PMA for 24 hr followed by 24 hr of normal media. (**G**) Quantification of experiment described in F. Data is represented as histograms of the sum of LGR5 intensity per cell. Gates show percent of high and low LGR5 +expressing cells. Analysis of 5014–11282 cells is shown. Asterisks represent significance from Mann Whitney analysis, ****=p<0.0001 etc. Scale bars represent 100 μM (*Figure 6—source data 5*).

The online version of this article includes the following video, source data, and figure supplement(s) for figure 6:

**Source data 1.** Source data for *Figure 6B*.

**Source data 2.** Source data for *Figure 6D*.

**Source data 3.** Source data for *Figure 6E*_top right.

**Source data 4.** Source data for *Figure 6E*_bottom right.

**Source data 5.** Source data for *Figure 6G*.

*Figure 6 continued on next page*

*Figure 6 continued*

**Figure supplement 1.** Pyrvinium Inhibits Wnt Signaling in Organoid Monolayers.

**Figure supplement 2.** Quantification of MYC Transcript Levels Following Treatment with Either 100 nM PMA (RED) or 1 μM PD0325901 (MEKi-GREEN).

**Figure 6—video 1.** ERK-KTR is Activated by Wnt Deprivation.

https://elifesciences.org/articles/78837/figures#fig6video1

**Figure 6—video 2.** ERK-KTR is Activated by Wnt Deprivation.

https://elifesciences.org/articles/78837/figures#fig6video2

cells. This process is an efficient way for the colon to maintain the correct proportions of stem and differentiated cells in distinct spatial compartments.

## Organoid monolayer system

We have shown that primary human colonic tissue can be expanded and transformed into mono-layers that phenocopy key characteristics observed in the tissue of origin. Although the immune and stromal compartments are not included in this culture method, different compartments of stem and

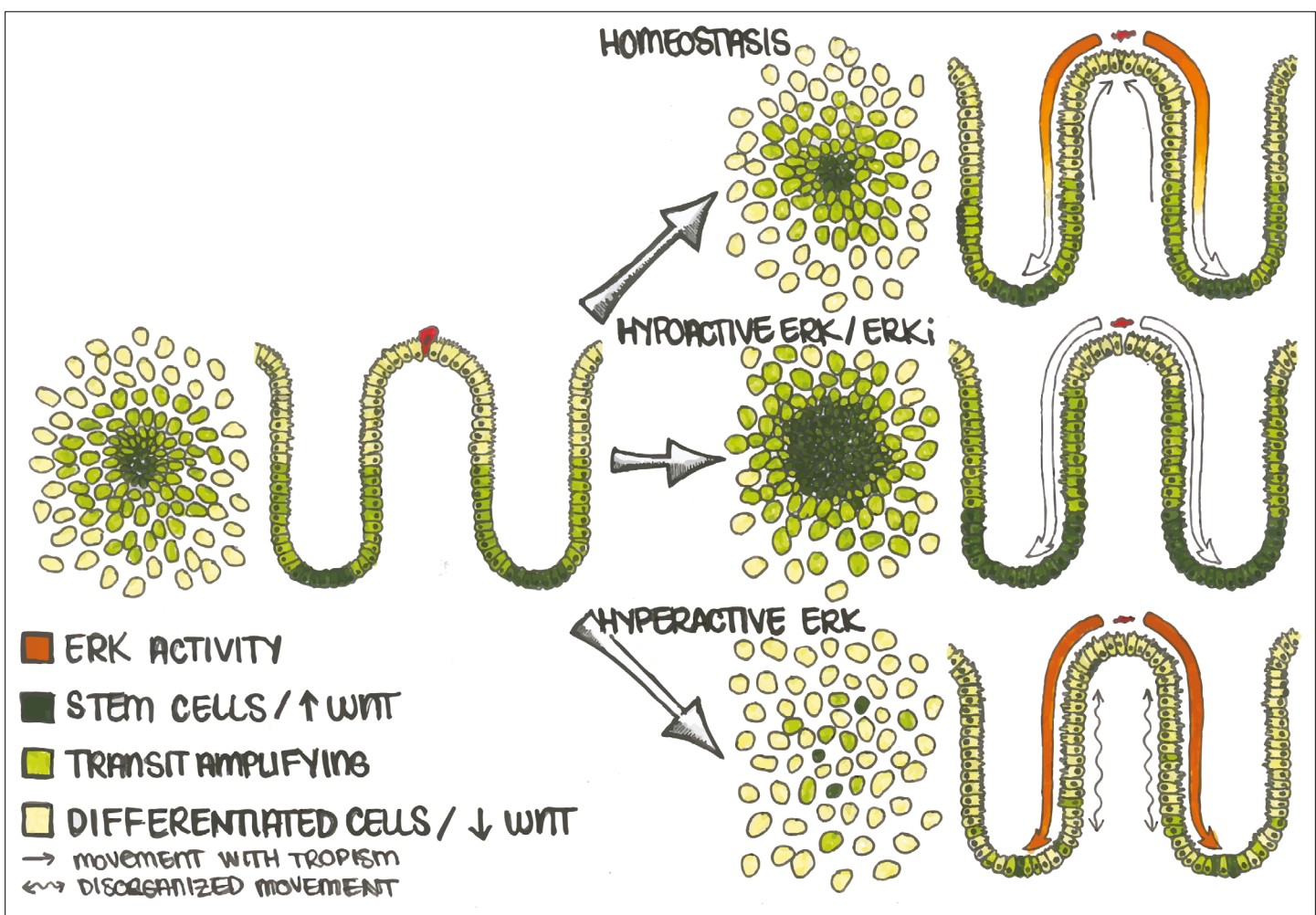

**Figure 7.** Apoptotic Cells Induce ERK Waves Which Limit the Stem Cell Compartment to a Defined Region. In the human colon, differentiated cells (light green) eventually undergo cell death (red cell) and are replaced by stem (dark green) and transit amplifying (lime green) cells. During homeostasis (top), apoptosis in the differentiated cell region triggers an ERK wave which propagates across the epithelium. This wave both triggers nearby cells to migrate (straight arrows) towards the site of apoptosis to maintain barrier function and also inhibits WNT signaling in order to maintain the correct proportion of stem, transit-amplifying, and differentiated cells. If ERK is inhibited (middle), the stem cell compartment expands and directional migration is lost. Conversely, if ERK is hyperactivated (bottom), cell movement becomes disorganized (wavy arrows) and spatially distinct cellular compartments are lost.

differentiated cells can be established and form unique physical properties depending on the source of the tissue (*Figure 1* and S1). The role of the stromal and immune compartments in the maintenance of tissue patterning and their effect on ERK dynamics is an exciting area of future study. The epigenetic modifications have been highlighted as a major mechanism that dictates the memory of stem cells from their tissue of origin (*Kim et al., 2010*). We hypothesize that dynamic behavior in critical regulatory kinases like ERK can also play an important role in adult epithelial tissues in the continued maintenance of cell differentiation and tissue architecture of preprogrammed stem cells.

## Potential self-organization mechanisms

Although our system is self-organizing, this work focuses more on the maintenance of stem and differentiated cell compartments in space. One future step is to define the mechanism by which organoid monolayers can establish these compartments from dissociated single cells. Recent work has highlighted mechanically mediated transcription factors and substrate stiffness as critical mediators of pattern formation in murine small intestine (*Serra et al., 2019*; *Gjorevski et al., 2022*; *Pérez-González et al., 2021*). Dissociation of 3D organoids into single cells most likely also triggers a wound healing response resulting in the production of fetal-like wound-associated epithelial cells (*Seno et al., 2009*; *Miyoshi and Stappenbeck, 2013*; *Miyoshi et al., 2017*; *Yui et al., 2018*; *Nusse et al., 2018*). These cells would aid in monolayer formation during self-organization (*Figure 1E*) then likely disappear as compartments are established. In the future, an air liquid interface culture with stromal cells present would limit wound healing and hypoxia phenotypes as described previously (*Wang et al., 2019*). However, the expression of localized Wnt target genes *LGR5* and *MYC* (*Figure 2B*) suggests that a fetal-like state has been repressed as the organoid monolayers mature and form patterns (*Yui et al., 2018*; *Nusse et al., 2018*).

## Apoptosis-induced survival

Previous work highlighted that apoptosis-induced ERK waves limit cell death to maintain epithelial integrity in a process termed apoptosis-induced survival (*Gagliardi et al., 2020*; *Valon et al., 2020*). In our organoid monolayer system, dying cells also triggered ERK waves, which directed cell movement toward the site of cell death (*Figure 3*). Although we did not calculate the probability of cell death following an ERK wave event, we observed a regular spacing of apoptotic cells, thus supporting the death refractory zone created by ERK waves described by others (*Gagliardi et al., 2020*; *Valon et al., 2020*).

## Colon architecture

ERK waves have been highlighted recently as central drivers of tissue architecture. The cochlear duct, which relies on its curved tissue architecture to function, is shaped by ERK waves during development (*Ishii et al., 2021*). ERK waves are also essential for proper regeneration of zebrafish scales (*De Simone et al., 2021*). Similarly, here we show that the spacing of stem cells in colonic organoid monolayers is correlated with the size of differentiation-promoting ERK waves. As extruded cells induce these ERK waves, they carve out a differentiated compartment, which drives cells towards apoptosis, resulting in positive feedback. These signals also limit stem cells, providing an elegant mechanism for separating regions of cell death from the stem cell compartment. However, we cannot say definitively that apoptosis instructs tissue patterning as caspase inhibition does not prevent ERK waves (presumably due to MMP-driven shedding of EGFR ligands being upstream of caspase activation) (*Gagliardi et al., 2020*). Optogenetic tools would need to be used to determine if tissue architecture can be controlled directly by ERK wave size, pattern, and duration.

## Mechanotransduction and ERK waves

Upstream of ERK waves, activation of rho-associated kinase (ROCK), and myosin light chain kinase (MLCK) are essential for apoptosis-driven membrane blebbing (*Coleman et al., 2001*; *Mills et al., 1998*; *Marchiando et al., 2011*) and extrusion (*Le et al., 2021*). Downstream of ERK waves, cells hit with a wave activate ROCK, driving cell migration and ultimately negative mechanical feedback as cells complete gap closure (*Hino et al., 2020*). We did not investigate the role of mechanical signaling in the maintenance of stem cell compartments using our system, but we think this is an exciting area.

It would be interesting to know if inhibition of ROCK or MLCK would induce or prevent waves, as both kinases play a role in activation and deactivation of paracrine signaling in this context.

## Maintaining a balance between stemness and differentiation

Here we see that acutely inducing ERK signaling across all cells destroys the stem cell niche, suggesting this mechanical feedback is essential to maintain stem cell homeostasis. Conversely, inhibition of the ERK pathway unleashes stem cells, resulting in aberrant expansion of the stem cell compartment. This suggests that the dynamics and location of ERK signals help to maintain the balance between stemness and differentiation. Recent studies in animal models support a mutually inhibitory relationship between the Wnt and ERK pathways in the normal gut. In the murine small intestine, the ERK activity is low in the crypts of Lieberkühn and higher as cells move out of the crypt. Inhibition of paracrine Wnt signaling results in ERK hyperactivation in the crypt and loss of LGR5+ stem cells (*Kabiri et al., 2018*), suggesting that Wnt is suppressing ERK signaling as it maintains stemness. Inversely, if ERK1/2 is knocked out at embryonic stages in mice, stem cells within the crypt proliferate and expand dramatically, displaying a ~twofold increase in LGR5 and OLFM4 (*Wei et al., 2020*). This phenotype is strikingly similar to what we have observed after chemical inhibition of the ERK pathway in our organoid monolayers (*Figure 6F*). In murine systems, oncogenic activation of ERK driver genes leads to decreases in the stem cell population (*Riemer et al., 2015*; *Leach et al., 2021*; *Reischmann et al., 2020*), which is similar to what we observed using an inducible KRAS$^{G12V}$ allele (*Figure 5*). During hair follicle development, a well-established patterning model system, EGFR is essential and serves to limit cell proliferation and stem cell numbers through attenuation of Wnt signaling (*Tripurani et al., 2018*). β-catenin is also essential for follicle formation, and hyperactivation of the Wnt pathway results in de novo follicle expansion (*Närhi et al., 2008*). Together, these studies support our hypothesis that these two pathways balance one another to maintain tissue homeostasis. ERK was activated via four separate methods in GiLA1 monolayers to assess its role in patterning maintenance: (1) PMA treatment (2) Expression of oncogenic KRAS (3) Removal of Wnt 3 a (4) Inhibition of Wnt via pyrvinium. Although activation of ERK occurred within hours in all four conditions, only PMA triggered rapid node loss, whereas other methods took days to induce patterning defects. We suspect this is due to the role of PKC (activated by PMA) in cell migration. PMA treatment most likely induced ERK activity and migration, leading to a previously described positive feedback loop (*Hino et al., 2020*). Our organoid monolayers lack stromal and immune cells, suggesting that the maintenance of patterning is most likely driven by signaling between epithelial cells rather than stromal or immune cell interactions with the colonic epithelium.

## Implications in human disease

The Wnt and ERK pathways are commonly hyperactivated during the development of colon cancer. This work unveils a relationship between these two pathways that maintains homeostasis in precancerous tissue. The aggressiveness of colon cancer is ranked in part by the loss of patterning that has occurred within the tissue (*Chang et al., 2014*; *Vogelstein et al., 1988*). Our work highlights the ERK and Wnt pathways as maintainers of patterning and therefore restrictors of dysplasia. One hypothesis is that Wnt pathway activation results in the suppression of ERK during early colon tumor development. Hyperactivating mutations such as oncogenic KRAS could be a selective response to Wnt activation. The result would be both the Wnt and KRAS pathways, which normally oppose one another, becoming hyperactivated together. This mutational combination, also requiring p53 suppression (*Serrano et al., 1997*), is seen in ~30% of colon tumors and results in highly mobile, proliferative, and adaptive clones.

The mutation frequency in the normal epithelium is far too high for DNA repair alone to explain how humans prevent the expansion of cancerous cells over a lifetime (*Rozhok and DeGregori, 2015*). How does normal tissue prevent the takeover of hyperproliferative clones throughout life? Pathways highlighted as tumor-driving can also play a vital role in tumor prevention within normal tissue. In human cell lines, oncogenic mutations that trigger sustained ERK signaling result in extrusion carried out by normal neighbors in cell mixing experiments. This surveillance, mediated by normal cells, is driven by the ERK waves (*Aikin et al., 2020*) and is an example of normal cells using a pathway described regularly as oncogenic to prevent the establishment of transformed clones. Similarly, epithelial cells rely on Wnt signaling to extrude precancerous cells that have acquired Wnt pathway activating mutations

(*Brown et al., 2017*). Our work highlights how Wnt and ERK can regulate the size and spacing of cellular compartments in pre-cancerous tissue and shows how delicately these pathways are balanced. Future work monitoring ERK and other central signaling pathways across many tumor PDOs with defined genetic backgrounds will unveil how patterning and signaling between single cells are dysregulated during tumor progression.

## Methods

### Preparation of 3D organoids from patient biopsies

The normal or tumor tissues from endoscopic ultrasound-guided fine-needle aspiration biopsies (EUS FNAs), or core needle biopsies were collected from consented patients by the (Tissue Acquisition and Repository for Gastrointestinal and HEpaTic Systems [TARGHETS], IRB 1909985869) facility located in the Arizona Health Science Center. Primary colonic organoid cell lines were anonymized by the Tissue Acquisition and Cellular/Molecular Analysis Shared Resource (TACMASR) at the University of Arizona Cancer Center. TACMASR is an on-campus biorepository to procure, store and retrieve biospecimens in a form that is deidentified and protects the privacy of the donors and confidentiality of the data collected. The individuals from whom the cells originated were resection patients at Banner University Medical Center. All research participants in this proposal receive the cells with de-identified and anonymous labels that cannot trace back to the individual or their families from which they came. Thus, no one involved in this study can access the subject's identities. Therefore, the study is exempt from being considered human subject research. Biopsy tissues were transported in 50 ml conical tubes containing collection media (Advanced DMEM/F-12 [Invitrogen 12634028] supplemented with 2 mM GlutaMax, 10 mM HEPES, 0.25 mg/ml Amphotericin B, 10 mg/ml Gentamycin, 1% Kanamycin, N-2 media supplement [Invitrogen 17502048], B-27 Supplement Minus Vitamin A [Invitrogen 12587010], 1 mM N-Acetyl-L-cysteine [Sigma A9165], 10 nM Nicotinamide [Sigma Aldrich; #N0636], 2.5 µM CHIR99021 [Tocris-Fisher, 4423; Apexbio Technology, A3011], and 2.5 µM Thiazovivin. Tissues were then washed, minced, and cryopreserved in organoid freezing media 70% seeding media [see next section], supplemented with 20%FBS, 10% DMSO, and 2.5 µM Thiazovivin) at the BioDROids core facility, located in the University of Arizona Cancer Center. Frozen tissues were prepared into organoids by thawing tissue pieces, mincing, and incubation with 1 mg/ml collagenase type 3 in phosphate-buffered saline (PBS) on a shaker for 10–25 min at room temperature. Cells were then removed from collagenase, washed with PBS, embedded into 100% Matrigel, and cultured in seeding media for 7 days. After organoids became established, low passage aliquots were cryopreserved in freezing media or infected with lentiviral constructs.

### Colorectal cancer organoids from the patient-derived models repository (PDMR)

A set of 16 colorectal cancer organoid models were purchased from the National Cancer Institute Patient-Derived Models Repository (PDMR; NCI-Frederick, Frederick National Laboratory for Cancer Research, Frederick, MD; https://pdmr.cancer.gov/). The set were selected on the basis of microsatellite stable and *APC* mutation-negative status (seven organoids) and a set of microsatellite stable and *APC* mutation-positive organoids were matched to the former set on the basis of age, gender, location, and stage. Two organoids from microsatellite unstable tumors were also selected, one *APC* mutation-negative and the other *APC* mutation-positive. In the present study, *APC* mutation-negative organoids 555926–031 R-V1-organoid and 624824–186 R-V1-organoid, *APC* mutation-positive organoids 276233–004 R-V1-organoid, 451658–271 R-V1-organoid and 722911–139 R-V1-organoid, and the microsatellite unstable organoid 616215–338 R-V2-organoid were expanded in Matrigel and organoid media according to the PDMR's published protocols.

### Growth conditions of 3D and organoid monolayers

#### Complete LWRN media

Advanced DMEM/F-12 (Invitrogen 12634028) supplemented with 2 mM GlutaMax, 10 mM HEPES, N-2 media supplement (Invitrogen 17502048), B-27 Supplement Minus Vitamin A (Invitrogen 12587010), 1 mM N-Acetyl-L-cysteine (Sigma A9165), 2500 units/mL Penicillin and 2.5 mg/mL streptomycin (0.05 mg/mL), 50% L-WRN Conditioned Media (*Sugimoto et al., 2018*), 100 ng/ml huEGF

(R&D 236-EG-01M), 500 nM A 83–01 (Tocris-Fisher, 29-391-0; APexBio-Fisher, 501150476), 10 µM SB 202190 (Tocris-Fisher, 12-641-0),100µg/ml Primocin, and 10 mM Nicotinamide (Sigma Aldrich; #N0636).

### Seeding media
Complete LWRN media supplemented fresh with 2.5 µM CHIR99021 (Tocris-Fisher, 4423; Apexbio Technology, A3011) and 10 µM Y27632 (Tocris-Fisher; 125410).

## Lentiviral infection of 3D organoids
$3×10^6$ HEK 293T cells were seeded in a 10 cm plate and transfected the following day using 500 µL of opti-MEM, 30 µL Gene Juice (Sigma #70967), 5 ug lentiviral construct, 3.25 µg psPAX2 (addgene #12260), and 1.75 µg pMD2.G (addgene #12259). After 24 hr, fluorescence was assessed, and media was collected for 3 days followed by centrifugation and filtration using 0.45 µM syringe filter. Lentiviral media was then kept at 4°C before 100 × concentration using LentiX concentrator (Takara #631232) and resuspension of virus in organoid seeding media; viral media was then stored at –80°C for up to 3 months. For infection of organoids, $1×10^5$ primary epithelial cells were harvested from 3D domes, washed, and trypsinzed as described previously. Cells were then counted using a hemocytometer and 40 K organoid cells were plated in suspension onto a 48 well plate and diluted 1:1 in vial media containing 8 mg/ml polybrene. Next, organoids were placed at 37°C for 1 hr the before centrifugation at 600 g for 1 hr at 32°C. Organoids were then harvested, washed, and embedded into 90% Matrigel and cultured in seeding media for 24 hr. Cells were then grown in complete LWRN for 3 days before fluorescence was assessed and >50% infection was achieved. Media was aspirated and organoids were resuspended in 500 ul trypsin. After dissociation, cells were mixed 5–10 times using a p200 pipette tip and passed through a 100 µm filter to ensure isolation of single cells.

## Preparation of organoid monolayers from 3D cultures
SCREENSTAR 384-well black plates (Grenier #781866) were coated with ice cold Matrigel (CB40230C) diluted 1:40 in Serum-free Advanced-DMEM F12 media (SFM) for 1 hr. Monolayers were prepared using our previously described protocol (*Thorne et al., 2018*; *Sanman et al., 2020*). Briefly, cells were removed from 3D Matrigel and resuspended in ice cold SFM and washed three times using ice cold SFM. Organoids were then dissociated by resuspension in Trypsin +10 µM Y-27 for 4 min followed by quenching in DMEM supplemented with 10% FBS and washing. Coating media was removed from 384-well plates and organoids plated at 7000 cells/well after counting using hemocytometer and cultured for 24 hr. Seeding media was then removed and replaced with complete LWRN media and changed daily for 7 days until cultures became confluent and tissue patterning was observed.

## High content imaging of organoid monolayers
384-well plates were imaged with fluorescent microscopy on a Nikon Eclipse TI2 automated microscope. For quantification of wave dynamics, organoids were imaged every 10–30 min. Tracking and segmentation of single cells in time lapse images in *Figure 2* was performed using the MATLAB program p53Cinema (*Reyes et al., 2019*). Analysis of the extracted data was performed in MATLAB. For the ERK activity heat maps in *Figure 3*, images of the ERK-KTRmRuby2 and the H2B-iRFP670 were captured every 7 min for 24 hr. For each frame, individual nuclei were segmented using a MATLAB script developed in-house that identifies nuclei based on the H2B-iRFP670 images. The cytoplasmic ERK-KTRmRuby2 signal was obtained by creating a two pixel wide annulus surrounding the nucleus and averaging the intensity. Cells with active ERK were identified as having a mean ERK-KTR cytoplasmic signal ≥the mean ERK-KTR nuclear signal. The density-based spatial clustering of applications with noise algorithm (dbscan, MATLAB, 2021b) was performed on the centroids of ERK active cells to identify spatial clusters of cells with active ERK. A boundary was drawn around the spatial clusters using the boundary algorithm in MATLAB. The ERK positive regions of each frame were summed up to generate the heat map in MATLAB.

## Cell movement analysis

For the cell movement data in *Figure 3D–E*, we performed two separate analyses. In the middle panel of 3D and the example cells in 3E, we first measured the distance of each nucleus $i$ at timepoint $t$ ($dNuc_{i,t}$, from the nucleus of the apoptotic cell during the apoptotic event using the distance formula:

$$dNuc_{i,t} = \sqrt{\left(x_{i,t} - x_a\right)^2 - \left(y_{i,t} - y_a\right)^2}$$

where $x_{i,t}$, $y_{i,t}$ are the x, y coordinates of the centroid of nucleus $i$ at timepoint $t$, and $x_a$, $y_a$ are the x, y coordinates of the centroid of the apoptotic nucleus. We then calculate the change in distance for each nucleus $i$ at timepoint $t$ $\left(dNuc_{i,t}\right)$ relative to the distance of the nucleus from the apoptotic cell at the time of apoptosis $\left(dNuc_{i,a}\right)$ using the following formula:

$$Mov_{i,t} = \left(dNuc_{i,a} - dNuc_{i,t}\right)$$

thus, $Mov_{i,t}$ will be 0 at the time of apoptosis, and this value will be positive for any other timepoint where the nucleus is closer to apoptotic nuclei and negative if it moves further away than at the time of the death event.

For the right panel of *Figure 3D*, we depicted relative movement from frame to frame using the following formula:

$$\Delta Mov_{i,t} = Mov_{i,t} - Mov_{i,t-1}$$

thus, this value will be positive if the nucleus moves closer to the apoptotic nucleus relative to the previous frame and negative if it moves away from the apoptotic nucleus.

## Plasmids

1. pLentiPGK DEST H2B-iRFP670 was a gift from Markus Covert, addgene #90237 (*Nguyen et al., 2015*).
2. pLentiPGK BLASTDEST ERK-KTRmRuby2 was a gift from Markus Covert, addgene #90231 (*Nguyen et al., 2015*).
3. PGF-H2BeGFP was a gift from Mark Mercola, addgene #21210 (*Kita-Matsuo et al., 2009*).
4. pInducer10b-EGFP-KRAS$^{G12V}$ was a gift from Ji Luo, addgene plasmid # 164925 (*Carver et al., 2014*).
5. 7TGC (Top-GFP) was a gift from Roel Nusse, addgene plasmid #24304 (*Fuerer and Nusse, 2010*).

## Antibodies and dyes

Goat polyclonal anti-Muc2 (Santa Cruz Biotechnology, sc-23170)
Alexa Fluor 546 Phalloidin (Thermofisher, A12379)
Click-iT EdU Cell Proliferation Kit for Imaging, Alexa Fluor 488 dye (Thermofisher, C10337)
DAPI (Thermofisher, D21490)
Rat monoclonal anti-Ki67 (Thermofisher, 14-5698-80)
Rabbit polyclonal DCAMKL (Abcam, ab31704)
Goat anti-Rat IgG (H+L) Cross-Adsorbed Secondary Antibody, Alexa Fluor 488 (Thermofisher, A-11006)
Donkey anti-Goat IgG (H+L) Secondary Antibody [DyLight 488] (Pre-adsorbed) (Novus, NBP1-74824)
NucView Caspase-3 Enzyme (Biotium #10403)

## Fluorescent-assisted cell sorting (FACS)

After infection with lentiviral reporters, dual reporter organoids were derived. Organoids were dissociated into single cells and sorted for similar levels of expression for the desired reporter (see figure S1C for representation of FACS plots generated using FloJo V10). Cell sorting was performed using the BD FACS Aria III flow cytometer.

## scRNA FISH

Experiments were done using previously validated and custom RNA probes for target mRNA of interest using molecular instruments HCR RNA FISH protocol for mammalian cells (*Choi et al., 2010*). Briefly, monolayers were fixed in 4% PFA, permeabilized in 70% EtOH at –20°C, hybridized at 37°C, and amplified overnight at room temperature. Monolayers were then imaged using a Nikon CSU-W1-Sora Spinning Disk Confocal Microscope. See *Supplementary file 1* for complete probe sequences.

## Acknowledgements

We thank members of the Paek and Thorne labs for helpful comments and discussion. This work was supported by National Institutes of Health grants GM130864 (ALP), GM147128 (CAT), CA242914 (NAE), DK118563 (JLM), T32CA09213, CA23074, and Wellcome Trust WT223952/Z/21/Z (CAT). Tissue acquisition was supported by the TACMASR core at University of Arizona, grant P30 CA023074.

## Additional information

### Funding

| Funder | Grant reference number | Author |
|---|---|---|
| National Institutes of Health | GM130864 | Andrew L Paek |
| National Institutes of Health | GM147128 | Curtis A Thorne |
| National Institutes of Health | CA242914 | Nathan A Ellis |
| National Institutes of Health | DK118563 | Juanita L Merchant |
| Wellcome Trust | WT223952/Z/21/Z | Curtis A Thorne |
| National Institutes of Health | TA32CA09213 | Curtis A Thorne |
| National Institutes of Health | CA23074 | Curtis A Thorne |

The funders had no role in study design, data collection and interpretation, or the decision to submit the work for publication. For the purpose of Open Access, the authors have applied a CC BY public copyright license to any Author Accepted Manuscript version arising from this submission.

### Author contributions

Kelvin W Pond, Conceptualization, Resources, Data curation, Software, Formal analysis, Supervision, Funding acquisition, Validation, Investigation, Visualization, Methodology, Writing - original draft, Writing - review and editing; Julia M Morris, Data curation, Investigation, Methodology; Olga Alkhimenok, Formal analysis; Reeba P Varghese, Juanita L Merchant, Data curation, Methodology; Carly R Cabel, Formal analysis, Methodology; Nathan A Ellis, Investigation, Methodology; Jayati Chakrabarti, Resources, Investigation; Yana Zavros, Resources; Curtis A Thorne, Conceptualization, Resources, Supervision, Funding acquisition, Investigation, Writing - review and editing; Andrew L Paek, Conceptualization, Resources, Data curation, Software, Formal analysis, Supervision, Funding acquisition, Investigation, Writing - original draft, Writing - review and editing

### Author ORCIDs

Kelvin W Pond ⓘD http://orcid.org/0000-0003-0519-4679
Curtis A Thorne ⓘD http://orcid.org/0000-0002-8711-8292
Andrew L Paek ⓘD http://orcid.org/0000-0002-2835-8544

## Ethics

All primary colonic organoid cell lines to be used in this study have been anonymized by the Tissue Acquisition and Cellular/Molecular Analysis Shared Resource (TACMASR) at the University of Arizona Cancer Center. TACMASR is an on-campus biorepository to procure, store and retrieve biospecimens in a form that is deidentified and protects the privacy of the donors and confidentiality of the data collected. The individuals from whom the cells originated were resection patients at Banner University Medical Center. All research participants in this proposal receive the cells with de-identified and anonymous labels that cannot trace back to the individual or their families from which they came. Thus, no one involved in this study can access the subject's identities. Therefore, the study is exempt from being considered human subject research.

This study was performed in strict accordance with the recommendations in the Guide for the Care and Use of Laboratory Animals of the National Institutes of Health. All of the animals were handled according to approved institutional animal care and use committee (IACUC) protocols (#2021-0772) of the University of Arizona.

## Decision letter and Author response

Decision letter https://doi.org/10.7554/eLife.78837.sa1
Author response https://doi.org/10.7554/eLife.78837.sa2

---

# Additional files

## Supplementary files

• Supplementary file 1. scRNA FISH custom probe sequences used for KRT20, MYC, and KRT20 are shown.

• Transparent reporting form

• Source data 1. Technical and Biological Replicates Performed.

## Data availability

All data generated or analyzed during this study are included in the manuscript and supporting file; Source Data files have been provided for all main figures.

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
