## [Editor Report]

This work employs timelapse microscopy to study behavior of human colonic organoids in monolayers as the organoids initially self-organize. The authors then follow maintenance of organization into densely clustered nodes that have increased cells in cell cycle and the remaining more sparsely populated regions with fewer cycling cells. This study builds on a literature demonstrating roles for signaling pathways like ERK in epithelial patterning that have been examined in the cell competition field and, more specifically, in mouse intestinal organoids. This manuscript should be relevant to a broad readership interested in how epithelial organoids can self-organize and the role of specific signaling pathways in the process. In addition, the technical aspects of the work with live human monolayer cultures observed over timelapse with extensive quantification of cell behavior represent a useful advance in the field.

---

## [Decision Letter]

**Decision letter after peer review:**

Thank you for submitting your article "Live-Cell Imaging in Human Colonic Monolayers Reveals Erk Waves Limit the Stem Cell Compartment to Maintain Epithelial Homeostasis" for consideration by *eLife*. Your article has been reviewed by 3 peer reviewers, one of whom is a member of our Board of Reviewing Editors, and the evaluation has been overseen by Didier Stainier as the Senior Editor. The reviewers have opted to remain anonymous.

Essential Revisions to make the paper suitable for *eLife* will include a response to the reviewer critiques and further examination of additional signaling pathways, in particular the Wnt pathway.

*Reviewer #1 (Recommendations for the authors):*

– The methods say "Cytoplasmic regions were defined by a donut-shaped ring _two pixels wide.." Is this supposed to read "at least 2 pixels wide"? Otherwise, it isn't clear how this is done, as cytoplasms would be greater than 2 pixels.

– Perhaps this should be in the public comments, but it seems to me in some of the sequences (most notably Video 3a) that the initial event (prior to caspase 3 positivity) is an expansion of the soon-to-be-apoptotic cell with actual outward movement of surrounding cells and then (pertinent to point 1 in the public comment) the expected actomyosin contraction that appears to bring the neighboring cells in tight as much of the apoptosing cell body rises up above the monolayer (as can be seen in brightfield). Yet, there is essentially no blue in the movement heatmap in Figure 3D. How can cells always move towards an apoptotic cell and never away (even randomly)?

– The PIV map in Figure S5 might be moved into main figures, as without some aid, readers may struggle to understand how cell motility is turned into the graphic representation. Indeed, Figure 3 is confusing until the supplemental videos are watched and all the supplemental data viewed. It would be great, if some sort of cartoon or guide to how movement and wave work could be provided here, or at least, say an explanation of what a node and what movements are measured can be shown in Fig, 3C. I spent a long time looking at individual cell clusters to see their relative positions over time to begin to understand the data.

– What is Figure 4H relative to? Is this in inhibitor experiments or just control?

– It is unclear what Figure 5F represents? What is "Sum/Cell"? Again, as above, differences in PMA-treated cultures may be due to cell death.

– There seemed to be scale bars in some of the videos, but I didn't see where their magnitude was given.

*Reviewer #2 (Recommendations for the authors):*

It's hard to find information on the number of independent biological organoid lines used in this study. It's important that this information be included. A cell shouldn't be considered a replicate.

Figure S4, can the concentrations be included? Was a dose-response curve used to determine the concentrations?

The link between apoptosis and tissue patterning proposed in the second paragraph of the "Erk waves instruct pattern" isn't supported by data. A disruption of apoptosis would need to be performed in order to make such a claim.

In the discussion, the "Organoid Monolayer System" section suggests that there may be a role for Erk signaling in tissue origin memory. This is not well supported by literature and ignores a lot of work on developmental patterning and epigenetic control of tissue identity including experiments by Nicole Le Dourain and other experimental embryologists that go back decades and would argue against a role for kinase signaling as a key to maintain tissue patterning.

*Reviewer #3 (Recommendations for the authors):*

1) The authors should provide a better characterization of the different cell types present in their organoid monolayers, such as Goblet cells, enteroendocrine cells, enterocytes, Tuft cells etc.

2) Is the intensity and duration of the Erk waves dependent on the amount of apoptosis?

3) It would be worth confirming these findings using the authors' tumor line in addition to their GiLA1 line. They could then ask whether the lack of node formation in their tumor line could be reversed, such as via the addition of ERK inhibitors, if those cells have a high Erk activation, as was suggested in the previous literature.

4) Since the authors argue that the Erk and Wnt pathway are mutually antagonistic, it would be informative to quantify Wnt activity during the Erk waves. This quantification could be performed with regard to space and time relative to the Erk waves. This experiment, while not essential for the current findings to have merit, would significantly help our understanding how those two pathways interact.

5) The -Wnt3a phenotype should be confirmed by using a Wnt3a inhibitor.

6) Figure 4B: please add a control image.

---

## [Author Response]

Reviewer #1 (Recommendations for the authors):– The methods say "Cytoplasmic regions were defined by a donut-shaped ring _two pixels wide.." Is this supposed to read "at least 2 pixels wide"? Otherwise, it isn't clear how this is done, as cytoplasms would be greater than 2 pixels.

Sorry for the confusion. It is true the cytoplasm is greater than two pixels. Yet, automated cytoplasmic segmentation often incorporates errors, especially in dense, confluent cultures. We found that a two pixel annulus adjacent to the nucleus was the most accurate cytoplasmic measurement. Since the reporter is diffuse in the nucleus the mean of this region is a good proxy for the concentration of the reporter in the cytoplasm. This approach is standard for KTR reporters and is used by other groups, in particular the developer of the ERK-KTR. For further details the following references describe this method in more detail:

Kudo, T., Jeknić, S., Macklin, D. et al. Live-cell measurements of kinase activity in single cells using translocation reporters. Nat Protoc 13, 155–169 (2018). https://doi.org/10.1038/nprot.2017.128

Regot S, Hughey JJ, Bajar BT, Carrasco S, Covert MW. High-sensitivity measurements of multiple kinase activities in live single cells. Cell. 2014 Jun 19;157(7):1724-34. doi: 10.1016/j.cell.2014.04.039.

– Perhaps this should be in the public comments, but it seems to me in some of the sequences (most notably Video 3a) that the initial event (prior to caspase 3 positivity) is an expansion of the soon-to-be-apoptotic cell with actual outward movement of surrounding cells and then (pertinent to point 1 in the public comment) the expected actomyosin contraction that appears to bring the neighboring cells in tight as much of the apoptosing cell body rises up above the monolayer (as can be seen in brightfield). Yet, there is essentially no blue in the movement heatmap in Figure 3D. How can cells always move towards an apoptotic cell and never away (even randomly)?

We agree with the reviewer’s description of the movement following the apoptotic event and have addressed their concerns by describing it more clearly in the Results section, (page 7 top) and adding a description of the different analyses in the methods section with formulas for our calculations (page 4). We apologize for not including this in the original submission.

To clarify, during the apoptotic event there is a slight movement away from the apoptotic cell, predominantly in the cells that are very close to the apoptotic cell. This slight movement away from the apoptotic cell is followed by movement towards the apoptotic cell, with the cells closest to the apoptotic cell moving before the cells that are further away. Importantly, since the movement away from the apoptotic cell is much smaller in magnitude than the subsequent movement towards the apoptotic cell it is hard to visualize in the heat maps. Therefore, we used two heat maps to depict cell movement following the apoptotic event.

In the middle heat map of Figure 3D, we measured the net movement of each cell toward the location of the apoptotic cell when it died. Since this measurement describes total distance traversed, a cell can be closer to the apoptotic cell in a given frame even if it migrated slightly further away in the current frame as compared to the previous frame. An example calculation would be as follows: if a nucleus was 20 μm away from the apoptotic nucleus at the time of death, and 10 μm away from the apoptotic cell at frame 11, this value would be 10 μm (20 μm – 10 um) for frame 11. If in the next frame, the nucleus was now 15 μm away the value would be 5 μm (20 μm – 15 um), even though it moved slightly further from the apoptotic nucleus. Thus, this calculation measures overall movement towards the apoptotic nucleus and does not capture the frame-to-frame movement.

To capture the frame-to-frame movement, we took the values described in the previous paragraph and subtracted frame t-1 from frame t. So, in the example described above, when the cell moved 5 μm away from the location of the apoptotic cell, this value would be -5uM, even if overall the cell has moved closer to the apoptotic cell since the time of the death event. This analysis is depicted in the heat map on the right in Figure 3D. You can see the movement away from the apoptotic cell in the cells closest to the apoptotic cell (top of the heat map). Also, before the apoptotic event the movements of each cell can be both away from the apoptotic cell and towards it, more or less at random as the reviewer suggested would occur.

– The PIV map in Figure S5 might be moved into main figures, as without some aid, readers may struggle to understand how cell motility is turned into the graphic representation. Indeed, Figure 3 is confusing until the supplemental videos are watched and all the supplemental data viewed. It would be great, if some sort of cartoon or guide to how movement and wave work could be provided here, or at least, say an explanation of what a node and what movements are measured can be shown in Fig, 3C. I spent a long time looking at individual cell clusters to see their relative positions over time to begin to understand the data.

Agreed. We have moved the PIV analysis to the main figures and have added a more detailed description of what is measured in the PIV analysis in the methods section and included a reference.

– What is Figure 4H relative to? Is this in inhibitor experiments or just control?

We agree this was not the best data representation. Figure 4H was showing the heterogeneity of wave size across all controls, no drug treatment was used. This figure was removed from the main figure and a more detailed analysis was performed and included in figure S5 in order to compare these distances directly to spacing of node regions.

– It is unclear what Figure 5F represents? What is "Sum/Cell"? Again, as above, differences in PMA-treated cultures may be due to cell death.

Thank you for pointing this out. We have relabeled it LGR5 intensity/Cell. This measurement is made by first segmenting individual cells and quantifying the LGR5 mRNA FISH intensity in the individual cell ROIs. For this reason, effects on cell death would not confound this data. The histogram is a minimum of 5000 single cell measurements showing the distribution of the RNA levels under the different treatments.

– There seemed to be scale bars in some of the videos, but I didn't see where their magnitude was given.

We thank the reviewer for catching this. The magnitude of the scale bars is 100 microns. This has been added to each of the video legends. Figure 5 was re-made and a 100-micron scale bar was added.

Reviewer #2 (Recommendations for the authors):It's hard to find information on the number of independent biological organoid lines used in this study. It's important that this information be included. A cell shouldn't be considered a replicate.

Agreed. We used a single patient derived organoid line for the majority of this study, yet we did compare to other patient derived lines that are normal and tumor in origin. We have added this information in the first paragraph of the Results section. We have also added a supplemental figure (Supplemental figure 1) comparing five different tumor lines to the Gila1 line to show patterning differences. We added an additional representative image of a normal patient derived monolayer to figure S1. In all experiments, technical replicates (multiple wells) and biological replicates (separate experiments) were performed. In some cases, the data is shown as each data point representing a single cell in order to show the distribution of response across single cells. The details for this can be found in the excel sheet provided with the exact number of replicates for each experiment. We also have added greater detail to the figure legends for number of replicates performed.

Figure S4, can the concentrations be included? Was a dose-response curve used to determine the concentrations?

A concentration of 1uM was used for each of the drugs. This was to show as was shown previously, that the KTR will shut off in response to well-established small molecule inhibitors. Minimal cell death was observed in the first 24 hours and cells showed clear inhibition of the KTR activity, so dose responses were not performed.

The link between apoptosis and tissue patterning proposed in the second paragraph of the "Erk waves instruct pattern" isn't supported by data. A disruption of apoptosis would need to be performed in order to make such a claim.

This is an excellent point and indeed it was previously shown that caspase inhibitors do not suppress Erk waves (shown in Gagliardi 2021, presumably due to MMP-driven shedding of EGFR ligands being upstream of caspase activation). Though we do believe that apoptotic events will influence patterning more detailed experiments potentially using optogenetic tools are needed to address this question directly. We have toned down the wording of this portion of the discussion to address this concern. Instead, we focus on the role of Erk waves in controlling patterning by replacing “apoptosis” with Erk waves in several places.

In the discussion, the "Organoid Monolayer System" section suggests that there may be a role for Erk signaling in tissue origin memory. This is not well supported by literature and ignores a lot of work on developmental patterning and epigenetic control of tissue identity including experiments by Nicole Le Dourain and other experimental embryologists that go back decades and would argue against a role for kinase signaling as a key to maintain tissue patterning.

We agree. Our intention was not to ignore well-established fate regulation mechanisms such as histone modifications at promoters and enhancers or the beautiful early lineage experiments of LeDourain and colleagues. Our work here simply suggests that Erk waves may help regulate cell plasticity and the architecture of the colon *during* homeostasis. It would be very interesting to see if epigenetic modifications were maintained after cells stopped producing Lgr5 for example in response to Erk activation and to detail their ability to reprogram back into stem cells, this is an area we are actively exploring. To address this concern, we have toned this section down and highlighted epigenetic modifications as the current major driver of tissue origin memory.

Reviewer #3 (Recommendations for the authors):1) The authors should provide a better characterization of the different cell types present in their organoid monolayers, such as Goblet cells, enteroendocrine cells, enterocytes, Tuft cells etc.

Agreed. We have added the percentages of different cell types in the Gila1 monolayers (Figure S1 E and F). We quantified stem cells, proliferative cells, colonocytes, tuft cells, and goblet cells. The GiLA1 line showed relatively low levels of goblet cells and slightly higher levels of stem cells, presumably due to its origin from an adenoma.

2) Is the intensity and duration of the Erk waves dependent on the amount of apoptosis?

Single waves were analyzed, all of which originated from a single death event. We found that if multiple cells died or a tear was formed in the monolayer, waves still occurred. We did not observe enough multiple cell deaths to quantify, presumably due to the anti-apoptotic nature of the Erk waves (Gagliardi 2021, Valon 2021). We did observe waves colliding with one another, but this did not change the size or duration of the Erk activity in each cell. This could be due to a maximal signal threshold being reached, or a limitation of the KTR reporter’s dynamic range.

3) It would be worth confirming these findings using the authors' tumor line in addition to their GiLA1 line. They could then ask whether the lack of node formation in their tumor line could be reversed, such as via the addition of ERK inhibitors, if those cells have a high Erk activation, as was suggested in the previous literature.

We agree this would be a fascinating area to study. We addressed this issue by testing the effect of inducible KRAS on the patterning of the monolayers (new Figure 5). We attempted to rescue patterning in the KRAS mutant organoids with a Mek inhibitor, but found that the KRAS-induced organoid monolayers became hypersensitive to Mek inhibitor and saw an increase in cell death. We feel to fully address this observation, which is reminiscent of “oncogene addiction” requires a careful secondary study that would be beyond the scope of our work here.

4) Since the authors argue that the Erk and Wnt pathway are mutually antagonistic, it would be informative to quantify Wnt activity during the Erk waves. This quantification could be performed with regard to space and time relative to the Erk waves. This experiment, while not essential for the current findings to have merit, would significantly help our understanding how those two pathways interact.

We completely agree. We are working to build a GSK-3β KTR reporter to address this as there are no currently available Wnt reporters that can address cell dynamics. Unfortunately to date, we do not have conclusive experiments that our GSK-3 KTR is faithfully reporting on endogenous GSK-3 activity. We have used a GFP reporter driven by TCF/LEF and found this to respond to Wnt inhibition over multiple days, however the long half-life of GFP prevents accurate dynamics data from being collected within the short time scales of stochastic Erk waves. Adding a degradation sequence to GFP also does not work as the basal fluorescence intensity of GFP is too low for accurate measurements. Therefore, we cannot include the crosstalk between Erk and Wnt dynamics directly at this time. Regardless, we have strengthened the effects of Wnt on Erk signal with our response to question 5 below.

5) The -Wnt3a phenotype should be confirmed by using a Wnt3a inhibitor.

Agreed. We have added data from experiments using the Wnt inhibitor Pyrvinium. We show that inhibition of Wnt results in activation of Erk and loss of nodes in a dose dependent manner (Figure 5E).

6) Figure 4B: please add a control image.

Control images were added to this figure.